# Evaluation of the Virulence Potential of *Listeria monocytogenes* through the Characterization of the Truncated Forms of Internalin A

**DOI:** 10.3390/ijms241210141

**Published:** 2023-06-14

**Authors:** Giulia Magagna, Maria Gori, Valeria Russini, Veronica De Angelis, Elisa Spinelli, Virginia Filipello, Vito Massimo Tranquillo, Maria Laura De Marchis, Teresa Bossù, Clara Fappani, Elisabetta Tanzi, Guido Finazzi

**Affiliations:** 1Food Safety Department, Istituto Zooprofilattico Sperimentale della Lombardia e dell’Emilia Romagna (IZSLER), Via A. Bianchi 9, 25124 Brescia, Italy; giulia.magagna@izsler.it (G.M.); elisa.spinelli@izsler.it (E.S.); guido.finazzi@izsler.it (G.F.); 2Department of Health Sciences, Università degli Studi di Milano, 20133 Milan, Italy; maria.gori@unimi.it (M.G.); clara.fappani@unimi.it (C.F.); elisabetta.tanzi@unimi.it (E.T.); 3Coordinated Research Centre EpiSoMI, Università degli Studi di Milano, 20133 Milan, Italy; 4Food Microbiology Unit, Istituto Zooprofilattico Sperimentale del Lazio e della Toscana “M. Aleandri”, Via Appia Nuova, 1411, 00178 Rome, Italy; valeria.russini@izslt.it (V.R.); veronica.deangelis@izslt.it (V.D.A.); marialaura.demarchis@izslt.it (M.L.D.M.); teresa.bossu@izslt.it (T.B.); 5Programmazione dei Servizi e Controllo di Gestione, Istituto Zooprofilattico Sperimentale della Lombardia e dell’Emilia Romagna (IZSLER), Via A. Bianchi 9, 25124 Brescia, Italy; vito.tranquillo@izsler.it; 6Department of Clinical Sciences and Community Health, Università degli Studi di Milano, 20122 Milan, Italy

**Keywords:** *Listeria monocytogenes*, virulence, *inlA*, premature stop codon (PMSC), food, listeriosis, food-processing environment, Italy

## Abstract

*Listeria monocytogenes* is a widespread Gram-positive pathogenic bacterium that causes listeriosis, a rather rare but severe foodborne disease. Pregnant women, infants, the elderly, and immunocompromised individuals are considered particularly at risk. *L. monocytogenes* can contaminate food and food-processing environments. In particular, ready-to-eat (RTE) products are the most common source associated with listeriosis. *L. monocytogenes* virulence factors include internalin A (InlA), a surface protein known to facilitate bacterial uptake by human intestinal epithelial cells that express the E-cadherin receptor. Previous studies have demonstrated that the presence of premature stop codon (PMSC) mutations naturally occurring in *inlA* lead to the production of a truncated protein correlated with attenuate virulence. In this study, 849 *L. monocytogenes* isolates, collected from food, food-processing plants, and clinical cases in Italy, were typed and analyzed for the presence of PMSCs in the *inlA* gene using Sanger sequencing or whole-genome sequencing (WGS). PMSC mutations were found in 27% of the isolates, predominantly in those belonging to hypovirulent clones (ST9 and ST121). The presence of *inlA* PMSC mutations in food and environmental isolates was higher than that in clinical isolates. The results reveal the distribution of the virulence potential of *L. monocytogenes* circulating in Italy and could help to improve risk assessment approaches.

## 1. Introduction

*Listeria monocytogenes* is a facultative intracellular foodborne pathogen responsible for human listeriosis, which is a rare but potentially severe infection, with manifestations ranging from self-limited gastroenteritis in healthy individuals to sepsis, meningitis, encephalitis, miscarriage, and stillbirth in at-risk groups (i.e., young, old, pregnant, immunocompromised—YOPI) [1,2]. *L. monocytogenes* is a ubiquitous pathogen able to colonize different environments, including soil, water, food-processing plants, and animal intestinal tracts [3]. Therefore, several food products can be a source of *L. monocytogenes*; however, considering that *L. monocytogenes* is inactivated by high temperatures, ready-to-eat (RTE) products are frequently involved in listeriosis outbreaks [4,5].

According to the Center for Disease Control (CDC), in the USA, an estimated 1600 people contract listeriosis each year, with about 260 people dying from the disease [6]. In 2021, the EU reported 2183 confirmed invasive human cases of listeriosis, with 196 deaths, keeping a stable trend of confirmed listeriosis cases in 2017–2021 after a long period of an increasing trend. In particular, Italy reported 241 cases of human invasive listeriosis [7].

Molecular typing of *L. monocytogenes* has a key role in the detection of outbreaks and in the identification of the source of food contamination [8]. To date, the isolates cluster into at least four lineages (I, II, III, and IV), divided into thirteen serotypes [9]. Isolates of serotypes 1/2b and 4b, belonging to lineage I, are predominantly associated with human listeriosis. Lineage II isolates, mostly serotype 1/2a, are usually found in the environment, food-processing plants, and foodstuffs, but they have also been implicated in a number of major listeriosis outbreaks [10,11]. The isolates of lineages III and IV are rarely isolated and are mainly in animals [12].

As an intracellular pathogen, *L. monocytogenes* possesses an arsenal of virulence factors that allow for host cell invasion and elusion of the host immune response. Current surveillance schemes consider all strains of *L. monocytogenes* as equally pathogenic. However, a number of studies have suggested that the *L. monocytogenes* virulence pattern is heterogeneous, and *L. monocytogenes* strains include hypervirulent and hypovirulent clones [13].

After ingestion through contaminated food, *L. monocytogenes* reaches the intestinal tract, and it is able to cross the intestinal barrier due to the interaction between the cell host receptor E-cadherin and the bacterial surface protein InlA, which has a key role in the virulence potential of *L. monocytogenes*, and consequently, in systemic infections [13,14]. For instance, a number of studies have reported that mutations leading to premature stop codons (PMSCs) in the *inlA* gene (2400 bp), resulting in a truncated form of InlA, are more prevalent in environmental and food-associated isolates (30–45%) and poorly represented among clinical isolates (5%), suggesting attenuated pathogenicity [14,15,16,17]. These findings suggest that the InlA sequence might serve as a virulence marker of *L. monocytogenes*. Nowadays, 30 mutations leading to PMSCs have been reported [8,18,19] (Table 1).

To the best of our knowledge, limited information is available regarding the presence of PMSC in the *inlA* of *L. monocytogenes* isolates in food, food-processing environments, and clinical cases in Italy. The aims of this study were: (i) the molecular characterization of food, environmental, and clinical isolates of *L. monocytogenes* collected in Italy using Sanger sequencing or whole-genome sequencing (WGS); (ii) the assessment of PMSCs in *inlA* in order to evaluate the virulence potential of *L. monocytogenes* isolated from food, food plants, and clinical samples collected within the surveillance of invasive listeriosis in Italy.

## 2. Results

### 2.1. Isolates

During the period of 2013–2022, a total of 849 isolates of *L. monocytogenes* were typed and assessed for the presence of PMSC in the *inlA* gene. In particular, all clinical isolates (n = 379) were included in the study, while for food and food-processing environments, a subset of isolates from a total of 993 collected within the national surveillance plan for *L. monocytogenes* were selected to be as representative as possible of all sources, food origin, and the sequence type (ST); namely n = 389 were from food and n = 81 were from food-processing environments. Food-associated strains were isolated from meat, fish, dairy, vegetables, and other products (Figure 1). All of the samples were collected in Italy. In particular, 318 isolates were collected in Lombardy, while 531 were isolated in the Lazio and Tuscany regions.

### 2.2. Multi-Locus Sequence Typing (MLST)

MLST detected a total of 64 different sequence types (ST), including 1 new ST (ST2687) (Appendix A). The predominant STs identified included ST9 (13%), ST5 (10%), and ST1 (9%) (Figure 2). The STs recovered from all the sources (food, environmental, and clinical) were ST2, ST3, ST5, ST7, ST8, ST9, ST37, ST121, ST155, ST217, ST288, ST325, and ST451. Certain STs were uniquely recovered from food, mostly isolated from meat products. ST145 and ST489 were exclusively isolated from the environment, and ten STs were exclusively found in clinical isolates (ST21, ST26, ST54, ST120, ST177, ST191, ST200, ST431, ST511, and ST2080) (Figure 2).

In Figure 3, the associations among the ten most prevalent STs and sources (environmental and clinical) or food origins (dairy, fish, meat, vegetables, and other) are visualized using correspondence analysis (CA) in R. The graphical representation shows a strong correlation between ST9 and meat isolates, between ST325 and dairy and environmental isolates, and between ST1 and ST5 and clinical cases. A minor association between ST9 and vegetables was also observed.

Fifty-eight percent (n = 490) of the isolates belonged to lineage II, and 42% (n = 359) to lineage I. Among all the samples, 42.5% (n = 361) belonged to serotype 1/2a (lineage II), 13.8% (n = 117) to serotype 1/2c (lineage II), 22.6% (n = 192) to serotype 4b (lineage I), and 19.3% (n = 164) to serotype 1/2b (lineage I). The serotype information of the 1.8% (n = 15) of the isolates (ST206, ST207, ST386, ST397, ST412, ST511, ST1584, and ST2687) was unavailable. Considering the sources, serotypes 1/2a and 1/2c were mainly found in food samples, serotype 1/2a was also predominantly found in environmental samples, and serotypes 1/2a and 4b were prevalent in clinical isolates (Figure 4).

In Figure 5, the associations among the serotypes and sources are visualized using CA in R. In particular, the graphical representation shows a strong correlation between serotype 1/2c and food-associated isolates, and between serotypes 4b and 1/2b and clinical cases. On the contrary, serotype 1/2a did not exhibit a particular association with any specific source. This is in agreement with the frequent findings of this serotype in both food and environmental isolates but also in clinical cases.

### 2.3. inlA Sequencing

The results of the 2400 bp *inlA* gene sequence analysis are shown in Appendix A. The full sequencing of the *inlA* gene revealed the presence of eleven PMSC mutation types (PMSC 4, PMSC 5, PMSC 6, PMSC 8, PMSC 11, PMSC 12, PMSC 13, PMSC 19, PMSC 25, PMSC 26, and PMSC 29) in 27% (n = 228) of the *L. monocytogenes* isolates. Among the isolates harboring a PMSC, 76.8% (n = 175) were collected from food, 18.4% (n = 42) were environmental isolates, and 4.8% were isolated from clinical cases (n = 11). Figure 6 shows the minimum spanning tree based on the STs of the isolates divided per lineage (lineage I (a) and II (b)) considering the presence of the full-length and truncated forms of InlA and the PMSC type. The figure highlights that the totality of almost all isolates belonging to lineage I present the entire gene, while several lineage II STs harbor the truncated form with different types of PMSC. The percentage of isolates with a PMSC per ST is summarized in Table 2. Indeed, within lineage I, the STs with a truncated InlA are rare, and within the STs harboring a PMSC, the isolates with a truncated InlA are an exception (2–3%). On the contrary, considering lineage II, STs harboring PMSCs are more common, and when this is the case, approximately all (90–100%) of the isolates present the truncated form.

In clinical isolates, the sequencing of the *inlA* gene showed the presence of four PMSC mutation types: PMSC 6, PMSC 19, PMSC 26, and PMSC 29. The characteristics of 11 patients and the corresponding clinical isolates are summarized in Table 3. In particular, nine clinical isolates were collected from the blood of patients with sepsis, one (#2) from liquor, and one (#6) was obtained from the peritoneal fluid of a patient with peritonitis. The median age of the individuals was 73 years, and all of them had at least one risk factor for invasive listeriosis. For one case (#11), the outcome was fatal. The BIGSdb-*Lm* platform grouped the 11 isolates into 4 distinct sequence types (ST2, ST9, ST121, and ST325) belonging to 4 different clonal complexes (CC2, CC9, CC31, and CC121). The IRIDA-ARIES platform maintained at the Italian National Institute of Health (Istituto Superiore di Sanità) (https://irida.iss.it; accessed on 24 May 2023), which includes all *L. monocytogenes* genomic sequences isolated from clinical cases in Italy from 2002 to the present, performed cluster analysis on the distance matrix of the Core Genome MLST (cgMLST) profile of each sample with respect to those of all the other samples already present in the platform (the allele distance threshold was set at 4). The eleven clinical isolates with a truncated InlA were not found to belong to any cluster.

## 3. Discussion

In the surveillance of listeriosis, molecular typing methods are particularly important to trace outbreaks and for the prevention and control of the spread of the disease [30,31]. This study helped us to evaluate the epidemiology of certain *L. monocytogenes* strains currently circulating in Italy and to investigate their potential virulence by *inlA* gene sequencing.

In line with previous studies, serotypes 1/2a and 1/2c were predominantly found in food isolates, and serotype 4b was prevalent in clinical isolates (Figure 4 and Figure 5) [14]. The ST distribution varied among the different sources. According to the literature, the ST9 and ST121 strains, considered hypovirulent clones, have frequently been found to persist in food and related environments, especially in meat plants, while ST1 is associated with more invasive forms of listeriosis, in particular, with neonatal infections [32,33,34,35]. Indeed, in the present study, the STs more prevalent in food isolates were ST9, mainly found in meat products (Figure 3), and ST121, while ST1, ST5, ST6, and ST8 were mostly associated with clinical isolates [30]. Moreover, as shown in Figure 3, our findings are consistent with the literature regarding the prevalence of ST325 in food and in environmental isolates, and it is considered one of the *L. monocytogenes*-persistent molecular types in food-processing plants linked to dairy sources [36].

It is well known that *L. monocytogenes* strains may carry several virulence factors, which result in different pathogenicity levels [34]. Indeed, numerous studies have indicated that not all *L. monocytogenes* strains are equally associated with invasive disease for the carriage of a truncated form of InlA, produced by PMSC mutation in the *inlA* gene, which results in impaired virulence [11,34]. In fact, mounting evidence suggests that the presence of a PMSC mutation in the *inlA* gene is more frequent in food-originating *L. monocytogenes* strains than in clinical strains [14,37]. Previous findings showed that *inlA* PMSCs are prevalently found in lineage II isolates more than in lineage I isolates; consistently, in our results, 99% of strains carrying PMSC mutations belonged to lineage II [11].

In this study, eleven types of PMSC mutations were found. A total of 228 isolates presented PMSCs, of which 223 (98%) belonged to serotypes 1/2a and 1/2c, 1 belonged to serotype 1/2b, 1 to serotype 4b, and 3 isolates had no available information, and, in line with previous studies, the highest prevalence was found in food (77%) and environmental (18%) isolates [38,39,40].

The isolates with PMSC mutation belonged to ST2 (n = 1), ST9 (n = 100), ST31 (n = 5), ST121 (n = 56), ST193 (n = 1), ST199 (n = 2), ST224 (n = 1), ST325 (n = 51), ST580 (n = 6), and ST717 (n = 2) (recently assigned) and to the newly assigned ST2687 (n = 3). These results are justified by the literature evidence. In fact, it was previously reported that ST9, ST31, ST121, and ST325 (CC31), with a truncated InlA, are strongly adapted to survive in food and food plants, and they have a minor role in causing clinical cases, showing very low invasiveness [34,36,41,42,43,44]. In addition, using correspondence analysis (CA), Figure 3 shows a frequent association between STs and isolates harboring a truncated InlA, such as ST9 and ST325, and meat and dairy isolates, while ST1, rarely found in isolates with PMSCs, was associated with clinical isolates.

Consistent with the previous studies of Jacquet et al. [16] and Tamburro et al. [17], in our findings, the presence of PMSCs in InlA was detected in about 3% of the clinical strains [16,17]. Even if it is well known that a truncated InlA is typically correlated with virulence attenuation, particularly in immunocompromised individuals, infection with *L. monocytogenes* carrying a truncated InlA is not sufficient to prevent severe clinical manifestation and can also be lethal.

We described eleven invasive listeriosis cases caused by *L. monocytogenes* strains carrying a truncated InlA, with one known case resulting in death. Six strains were classified as ST9/CC9, and three as ST121/CC121, STs typically strongly associated with a food origin, underrepresented in clinical samples, and known to harbor truncated forms of InlA [13]. Notably, one case was classified as ST2/CC2, which has been previously described as a lineage I hypervirulent clone, responsible for severe clinical listeriosis. Lineage I isolates are historically linked to listeriosis cases and outbreaks and usually present a full *inlA*. Interestingly, in our dataset, another isolate belonging to lineage I ST224 harbored a truncated *inlA* (S1).

Moreover, it is noteworthy that six clinical isolates belonging to ST9 (serotype 1/2c) presented the PMSC mutation type 19. This result is consistent with the findings of Gelbíčová et al. [18] and Medeiros et al. [45], who have detected a PMSC in the same position in *L. monocytogenes* serotype 1/2c isolates from clinical samples [18,46]. Analysis of the *inlA* gene showed that the presence of PMSCs may interfere with strains’ invasion ability depending on their nucleotide position. In particular, the presence of PMSC 19 at nucleotide position 976 may not affect the capacity of *L. monocytogenes* strains to adhere to and invade epithelial cells [45]. For this reason, it is crucial not only to verify the presence or absence of *inlA* mutations but also the PMSC mutation type. The potential effect of each mutation on the formation and stability of the InlA and E-cadherin complex could be investigated with molecular modeling [46].

## 4. Materials and Methods

### 4.1. L. Monocytogenes Isolates

Food and food-processing plant isolates included in this study were selected from an ongoing monitoring plan for the characterization of *L. monocytogenes* or from foodborne diseases. Clinical isolates collected in Lombardy were selected by the Centre of Epidemiology and Molecular Surveillance of Infections (CRC EpiSoMI) of the University of Milan (UNIMI). The clinical and food/environment-associated isolate dataset from the Lazio and Tuscany regions included all the isolates sent from hospitals and private laboratories and received by the Regional Laboratory for Foodborne Human Pathogens (LRPTAU) and Regional Reference Centre for Pathogenic Enterobacteria (CREP) at the Food Microbiology Unit of IZSLT, central division of Rome, respectively.

### 4.2. Sanger Sequencing

#### 4.2.1. DNA Extraction from *L. monocytogenes* Isolates

From a total of 148 isolates, the genomic DNA was extracted by boiling. Briefly, cultures were grown overnight at 37 °C in blood agar plates. For each sample, a colony was added to 100 µL of Chelex^®^ (6%) (Sigma-Aldrich, St. Louis, MO, USA) and incubated at 56 °C for 15 min and then at 99 °C for 10 min. The samples were centrifuged at 13,000× *g* for 5 min. The supernatant was collected and stored at 4 °C until use.

#### 4.2.2. Multi-Locus Sequence Typing (MLST)

MLST was performed according to Ragon et al. [9]. Briefly, for each gene, PCR was carried out with a final volume of 25 µL: 10.9 µL of water, 12.50 µL of HotStarTaq Master Mix kit (2X) (Qiagen, Hilden, Germany), 0.30 µL of primer F (50 µM), 0.30 µL of primer R (50 µM) (Table 4), and 1 µL of DNA. The thermic profile was set to 10 min denaturation at 95 °C and 35 cycles of 30 s at 94 °C, 30 s at 52 °C, and 2 min at 72 °C, and a final extension step at 72 °C for 10 min for all genes except for *bglA*, which has an annealing temperature of 45 °C. The successful amplification of the PCR products was visualized with capillary electrophoresis on a QIAxcel Advanced System (Qiagen, Hilden, Germany) using the QIAxcel^®^ DNA Screening Kit (v2.0) (Qiagen, Hilden, Germany) with QX Alignment Marker 15 bp/1 kb (Qiagen, Hilden, Germany) and QX DNA Size Marker 50–800 bp v2.0 (5 ng/μL) (Qiagen, Hilden, Germany).

The PCR products were enzymatically purified with ExoSAP-IT™ Express PCR Product Cleanup Reagent (Thermo Fisher Scientific, Waltham, MA, USA) according to the instructions. Cycle sequencing was performed using 3 µL of DNase-RNase-free water, 2 µL of BigDye™ Terminator v1.1 Cycle Sequencing (Thermo Fisher Scientific, Waltham, MA, USA), 1 µL of BigDye^®^ Terminator v1.1 5X Sequencing Buffer (Thermo Fisher Scientific, Waltham, MA, USA), 2 µL of primer (1.6 µM), and 2 µL of PCR product. The plate was loaded in the GeneAmp^®^ PCR System 9700 (Thermo Fisher Scientific, Waltham, MA, USA), and the thermal cycling conditions consisted of a denaturation step at 96 °C for 1 min, 25 cycles of 10 s at 96 °C and 5 s at 50 °C, and a final extension step at 60 °C for 4 min. The products were purified using the BigDye Xterminator™ Purification Kit (Thermo Fisher Scientific, Waltham, MA, USA) according to the instructions, and the samples were sequenced on an Applied Biosystems Seqstudio Genetic Analyzer (Thermo Fisher Scientific, Waltham, MA, USA) with the medium_BDX run module. The consensus sequences for the seven genes were created and aligned with Molecular Evolutionary Genetics Analysis Version 6.0 (MEGA version 6) software [47]. The allele number and the ST were attributed using the MLST database of Institut Pasteur (http://bigsdb.pasteur.fr/listeria/listeria; accessed on 22 September 2022). The MLST data were visualized using minimum spanning trees (MSTs) generated by PHYLOViZ-2.0 software [48].

#### 4.2.3. *inlA* Sequencing

The *inlA* gene was sequenced for all isolates of *L. monocytogenes*. The gene was amplified by PCR using three pairs of primers that covered the whole *inlA* sequencing of 2400 bp. Two pairs of primers were previously detailed by Gelbíčová et al. [18]; the third pair of primers was newly designed. The *inlA* sequencing was performed according to Magagna et al. [49].

### 4.3. Whole-Genome Sequencing and In Silico Analysis

Among the 701 sequenced isolates, the genomic DNA of 170 isolates was extracted using the GenElute bacterial genomic DNA kit (Sigma-Aldrich, St. Louis, MO, USA), while the genomic DNA of 531 isolates was extracted with the automatic extraction system QIAsymphony (Qiagen, Hilden, Germany). Sequencing was conducted on a MiSeq system (Illumina, San Diego, CA, USA) using the MiSeq Reagent Kit v3 or v2 (2 × 300 or 2 × 250 bp), the Nextera XT DNA Sample Preparation kit, and the Nextera XT Index Kit, following the manufacturer’s instructions.

The raw read quality was assessed with FastQC (v0.11.5) and low-quality reads and adapters were trimmed using Trimmomatic (v0.39) using the following quality filter: a minimum quality of Q30, a window size of 10 with Q20 as the average quality, and a minimum length read of 50 bp [50,51]. The high-quality reads were assembled de novo into contigs using SPAdes (v3.13.0) with the careful option on, and contigs shorter than 500 bp were removed [52]. The assembly quality was assessed with QUAST (v5.0.2) [52,53,54].

For isolates from the Lazio and Tuscany regions, the detection of the *inlA* gene was performed using several tools. Firstly, the trimmed reads were analyzed with the tool VirulenceFinder (v.2.0) to detect the main virulence genes [55]. The sequences of the target gene were extracted from the resulting files of each sample, and then aligned and translated (Bacterial code, translate Table 11, available at https://www.ncbi.nlm.nih.gov/Taxonomy/taxonomyhome.html/index.cgi?chapter=cgencodes; accessed on 9 January 2023) with Geneious Prime^®^ (v2021.0.3) (Biomatters Ltd., Auckland, New Zealand). The alignment was visually checked for mutation or indels that produced a PMSC. For the in silico allele identification of the *inlA* gene, BLAST (v2.11.0) was performed on the *inlA* sequences extracted, using as a reference the *inlA* database retrieved from the BIGSdb-Lm (https://bigsdb.pasteur.fr/listeria/; accessed on 1 January 2022) [56,57]. In samples for which the previous method failed to retrieve the *inlA* sequence, the research was performed using BLAST (v2.11.0) directly on the assemblies [56].

For the Lombardy region clinical isolates, the STs, CCs, and the presence of PMSCs were determined in silico using the MLST database of Institut Pasteur (https://bigsdb.pasteur.fr/listeria/; accessed on 31 January 2023). When the MLST database reported the presence of a PMSC, the mutation position and the length of the resulting truncated InlA protein were confirmed by MEGA version 6 [36].

### 4.4. Statistical Analysis

To explore the relationships among the STs or serotypes and the source or food origin, the data were organized into contingency tables (CT) (Appendix A), and then each CT was analyzed using the correspondence analysis (CA) using CA() function of FactoMineR package in R language [58,59]. In order to interpret the distance between the column points and row points, the results of the CA are all presented graphically via an asymmetric biplot using the fviz_ca_biplot() function of the factoextra package [60]. In asymmetric biplots, if the angle between two arrows is acute, then there is a strong association between the corresponding row and column.

## 5. Conclusions

In this study, we report 11 clinical cases caused by strains harboring a PMSC and, therefore, generally considered hypovirulent, such as ST9 and ST121. Although isolates harboring a PMSC are correlated to attenuated pathogenicity and are not associated with epidemiological clusters, the number of clinical cases caused by such strains reported in this study is not negligible, indicating that the possibility of strains with a truncated InlA causing severe symptoms, and even fatalities, is not so rare. In addition, while historically PMSCs have widely been reported within lineage II, we give evidence of at least two strains belonging to lineage I and harboring a truncated inlA, one of them belonging to a well-known hypervirulent clone, namely, ST2.

These results underline the need to investigate the presence of PMSCs, but, above all, the *inlA* mutation type effects influencing the capability of *L. monocytogenes* strains to invade the host cells, given their possible role as a public health risk and causing severe symptoms in fragile individuals.

On a more general level, our findings are consistent with the literature showing that PMSCs are predominantly found in food and environmental *L. monocytogenes* isolates and are underrepresented among clinical isolates, which gives a picture of the distribution of *L. monocytogenes* types isolated from food, environmental, and clinical sources in Italy.

This information is fundamental for risk assessment purposes where updated descriptive data of the strains circulating in the country are required. Indeed, the data produced by this study will be also useful to improve risk analysis approaches and will serve as a basis for future works investigating the role of PMSCs in *inlA* and other virulence factors of this bacterial pathogen.

## Figures and Tables

**Figure 1 ijms-24-10141-f001:**
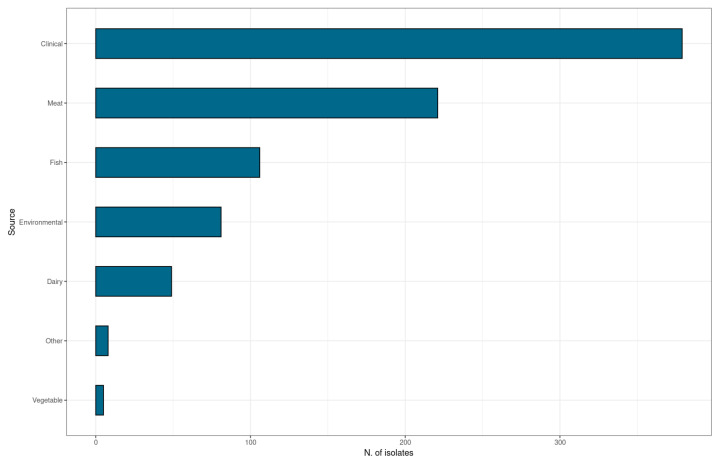
Source distribution of *Listeria monocytogenes* isolates.

**Figure 2 ijms-24-10141-f002:**
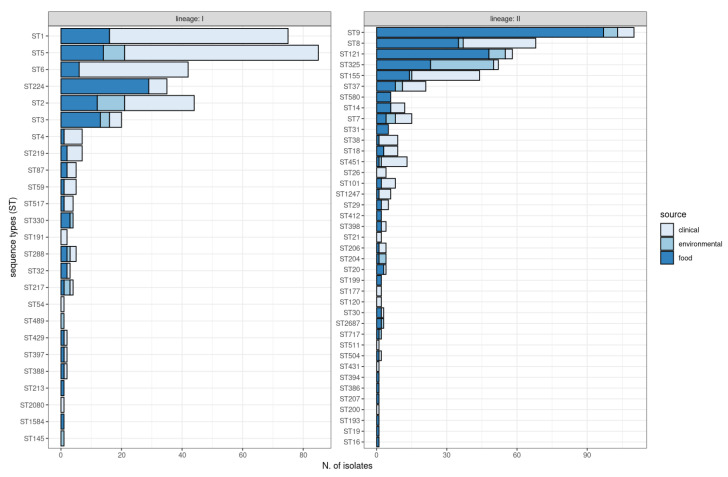
ST distribution and sources of *Listeria monocytogenes* isolates belonging to lineage I and lineage II.

**Figure 3 ijms-24-10141-f003:**
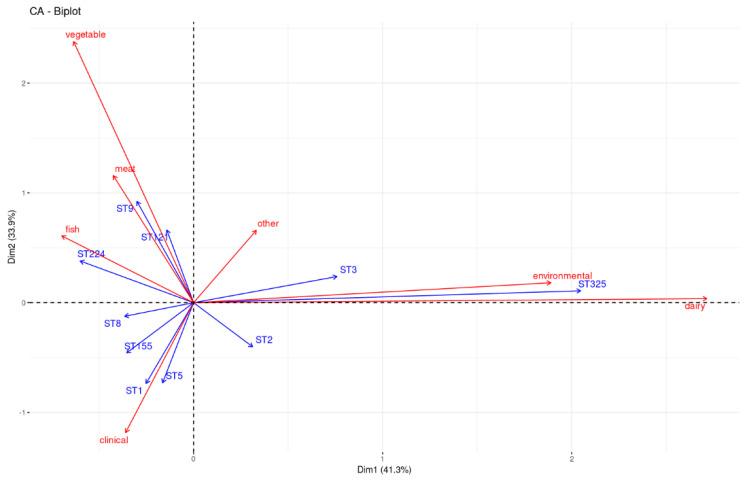
Correspondence analysis (CA) between the ten most prevalent STs (represented in blue) and sources or food origins (represented in red) of *Listeria monocytogenes* isolates. The more acute the angle between two arrows, the stronger the association between the corresponding ST and source or food origin.

**Figure 4 ijms-24-10141-f004:**
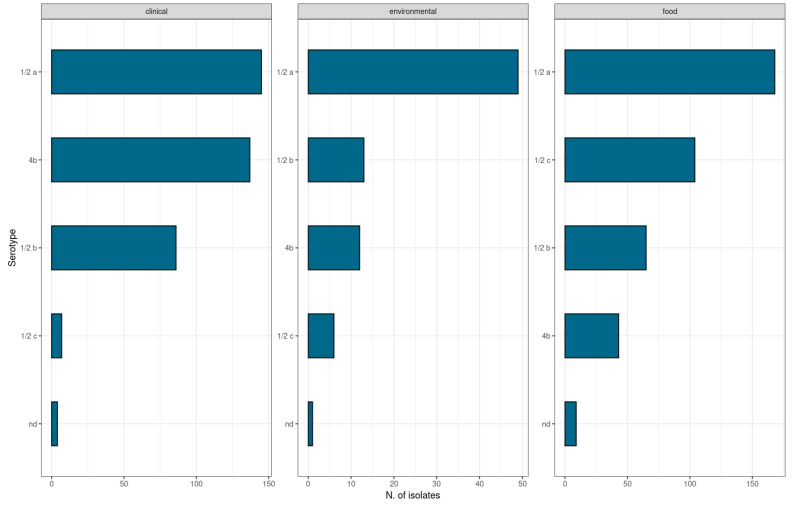
Prevalence of serotypes per source. nd = not determined. Serotype information was unavailable.

**Figure 5 ijms-24-10141-f005:**
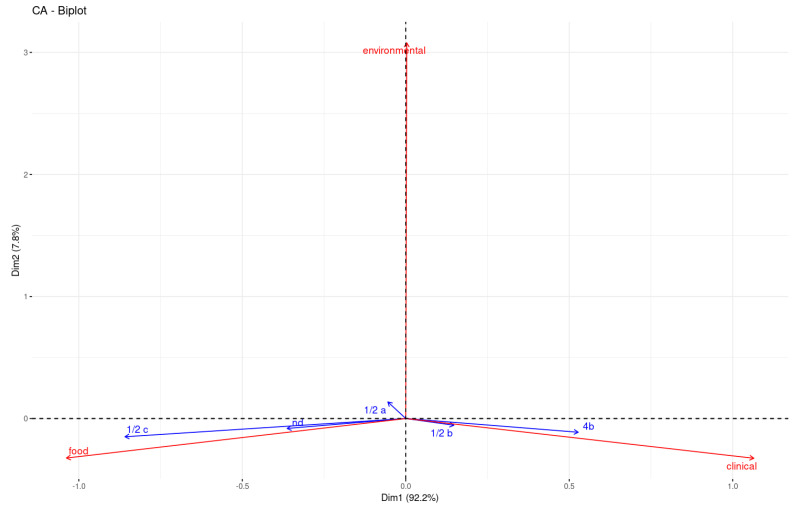
Correspondence analysis (CA) between serotypes (represented in blue) and sources (represented in red) of *Listeria monocytogenes* isolates. The presence of an acute angle between two arrows proves that there is a strong association between the corresponding serotype and source.

**Figure 6 ijms-24-10141-f006:**
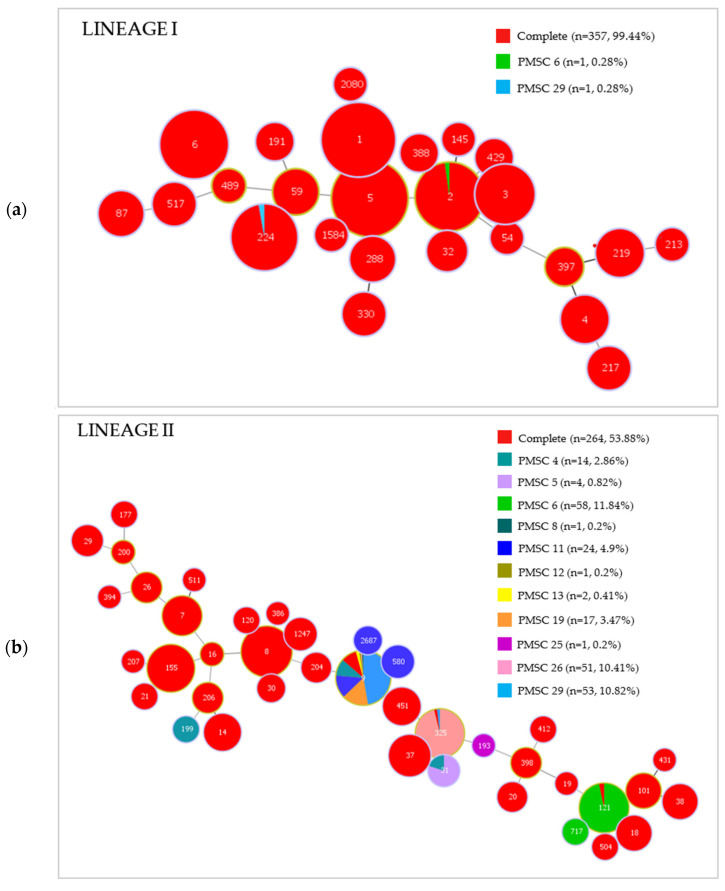
Minimum spanning tree of the 849 *Listeria monocytogenes* isolates belonging to lineage I (**a**) and lineage II (**b**) typed with MLST. Each circle represents a single ST indicated on the tree by the corresponding number. For each ST, isolates with full-length InlA and isolates harboring a truncated protein, and the type of mutation are represented by the colors in the legend.

**Table 1 ijms-24-10141-t001:** PMSC mutation types of *Listeria monocytogenes*.

PMSC Mutation Type	Nucleotide Position of Mutation	Length of Truncated InlA (aa)	Lineage	References
1	1818 (T→A)	605	I	Nightingale et al., 2005 [20]
2	1966 (C→T)	655	I	Nightingale et al., 2005 [20]
3	2100 (C→G)	699	II	Nightingale et al., 2005 [20]
4	12 (deletion A)	8	II	Felício et al., 2007 [21]
5	565 (C→T)	188	II	Van Stelten and Nightingale, 2008 [22]
6	1474 (C→T)	491	II	Olier et al., 2003 [23]
7	1684 (C→T)	561		Van Stelten and Nightingale, 2008 [22]
8	1380 (G→A)	459	II	Rousseaux et al., 2004 [24]
9	1540 (deletion G)	518		Rousseaux et al., 2004 [24]
10	1961 (insertion T)	676		Rousseaux et al., 2004 [24]
11	2054 (G→A)	684	II	Rousseaux et al., 2004 [24]
12	1637 (deletion A)	576		Jonquières et al., 1998 [25]
13	1579 (A→T)	526		Handa-Miya et al., 2007 [26]
14	1615 (C→T)	538	II	Ragon et al., 2008 [9]
15	229 (C→T)	76	II	Van Stelten et al., 2010 [27]
19	976 (G→T)	325	II	Gelbíčová et al., 2015 [18]
20	288 (C→A)	95	I	Moura et al., 2016 [19]
21	806 (T→A)	268	I	Moura et al., 2016 [19]
22	1756 (C→T)	585	I	Moura et al., 2016 [19]
23	1939 (A→T)	646	I	Moura et al., 2016 [19]
24	13 (C→T)	4	II	Moura et al., 2016 [19]
25	12 (deletion A)	25	II	Moura et al., 2016 [19]
26	277 (G→T)	92	II	Moura et al., 2016 [19]
27	576 (insertion T)	194	II	Moura et al., 2016 [19]
28	736–738 (CCA→TAG)	245	II	Moura et al., 2016 [19]
29	1635 (deletion A)	576	II	Moura et al., 2016 [19]
30	1741 (C→T)	580	II	Moura et al., 2016 [19]
31	2208 (deletion A)	753	I	Kurpas et al., 2020 [8]
32	1041 (C→A)	346	I	Tsai et al., 2022 [28]
33	937 (deletion C)	312		Ji et al., 2023 [29]

**Table 2 ijms-24-10141-t002:** Percentage of isolates harboring a PMSC divided per ST.

ST ^1^	Lineage	Internalin A	
		Complete	PMSC ^2^
ST2	I	97.7% (n = 43)	2.3% (n = 1)
ST9	II	9.1% (n = 10)	90.9% (n = 100)
ST31	II	0	100% (n = 5)
ST121	II	3.4% (n = 2)	96.6% (n = 56)
ST193	II	0	100% (n = 1)
ST199	II	0	100% (n = 2)
ST224	I	97.1% (n = 34)	2.9% (n = 1)
ST325	II	1.9% (n = 1)	98.1% (n = 51)
ST580	II	0	100% (n = 6)
ST717	II	0	100% (n = 2)
ST2687	II	0	100% (n = 3)

^1^ ST = sequence type; ^2^ PMSC = premature stop codon.

**Table 3 ijms-24-10141-t003:** Characteristics of truncated internalin A in clinical *Listeria monocytogenes* strains circulating in Italy, 2019–2021.

Patient	Age (Years)	Underlying Conditions	Outcome	Site of Infection	ST ^1^	CC ^2^	PMSC ^3^ Type
#1	99	Immunosuppressive drugs	Unknown	Blood	2	2	6
#2	73	Previous heart attack	Unknown	Liquor	9	9	19
#3	61	Unknown	Unknown	Blood	9	9	29
#4	74	Unknown	Unknown	Blood	9	9	29
#5	81	Unknown	Unknown	Blood	9	9	29
#6	87	Liver failure	Favorable	Peritoneal fluid	9	9	29
#7	67	Cancer	Favorable	Blood	9	9	19
#8	38	Unknown	Unknown	Blood	121	121	6
#9	71	Cancer	Favorable	Blood	121	121	6
#10	89	Immunosuppressive therapy	Unknown	Blood	121	121	6
#11	71	Cancer	Lethal	Blood	325	31	26

^1^ ST = sequence type; ^2^ CC = clonal complex; ^3^ PMSC = premature stop codon.

**Table 4 ijms-24-10141-t004:** Primers used for MLST PCR amplification.

Gene	Sequence
*abcZ*	abcZoF: GTTTTCCCAGTCACGACGTTGTATCGCTGCTGCCACTTTTATCCA abcZoR: TTGTGAGCGGATAACAATTTCTCAAGGTCGCCGTTTAGAG
*bglA*	bglAoF: GTTTTCCCAGTCACGACGTTGTAGCCGACTTTTTATGGGGTGGAG bglAoR: TTGTGAGCGGATAACAATTTCCGATTAAATACGGTGCGGACATA
*cat*	catoF: GTTTTCCCAGTCACGACGTTGTAATTGGCGCATTTTGATAGAGA catoR: TTGTGAGCGGATAACAATTTCAGATTGACGATTCCTGCTTTTG
*dapE*	dapEoF: GTTTTCCCAGTCACGACGTTGTACGACTAATGGGCATGAAGAACAAG dapEoR: TTGTGAGCGGATAACAATTTCATCGAACTATGGGCATTTTTACC
*dat*	datoF: GTTTTCCCAGTCACGACGTTGTAGAAAGAGAAGATGCCACAGTTGA datoR: TTGTGAGCGGATAACAATTTCTGCGTCCATAATACACCATCTTT
*ldh*	ldhoF: GTTTTCCCAGTCACGACGTTGTAGTATGATTGACATAGATAAAGA ldhoR: TTGTGAGCGGATAACAATTTCTATAAATGTCGTTCATACCAT
*lhkA*	lhkAoF: GTTTTCCCAGTCACGACGTTGTAAGAATGCCAACGACGAAACC lhkAoR: TTGTGAGCGGATAACAATTTCTGGGAAACATCAGCAATAAAC

## Data Availability

The data presented in this study are available upon request from the corresponding author.

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
