# Peer review of "Evaluation of the Virulence Potential of Listeria monocytogenes through the Characterization of the Truncated Forms of Internalin A"

_ijms, 2023, doi:10.3390/ijms241210141_

Round 1

Reviewer 1 Report

The authors present the molecular typing of L. monocytogenes observed in Italy over an extended period from both clinical cases and non-human sources, with a focus on the presence of PMSC in inlA in the isolates. The presence and biological/clinical significance of PMSC in this virulence factor has been described before several times, yet this manuscript shows comprehensive surveillance data from Italy.

As a general remark I would like to raise the point that although it seems from the Material&Method section that the majority of isolates have been analyses by WGS, the analysis in the manuscript remains limited to ST (and CC) level rather than the higher resolution (cgMLST) possible by this method. It might be interesting to use the full potential of this technique to elaborate more on certain finding (e.g demonstrated associations ST <-> PMSC-types, cfr particular comments below).

Specific comments:

Line 90: Which selection criteria where used to include the isolates to the study?

Figure 5: Elaborate more on the conclusions of the figure, more in particular the association of serotype 1/2a.

Figure 6: Shown without description of observations. Elaborate more on the findings or remove since this figure is redundant with Figure 2 (given the correlation between ST and serotype)

Paragraph 2.3 (line 158-166): Observed correlation between clonal complex and PMSC type in clinical isolates. It would be very useful to look at a higher resolution than cc among these isolates (cgMLST) to see whether these are part of a cluster.

Line 201: It would be interesting to indicate the percentage of strains with a PMSC per Sequence Type.

Line 223: Look for ST2/CC2 with this PMSC in public databases(BigsDB)?

Line 224: Table 2 indicates ST9 isolates with PMSC type 19/29, error?

In general a lot of complex, composite sentences.

Reviewer 2 Report

This manuscript by Magagna et al. has studied the truncated forms of Internalin A in Listeria monocytogenes isolates, collected from food, food processing plants, and clinical cases in Italy using Sanger sequencing and whole genome sequencing. Using distribution plots and correspondence analyses the authors report the higher occurrence of inlA PMSC mutations in food and environmental isolates as compared to clinical isolates.

 Comments:

1.    Since authors have whole genome sequencing data available, why did they focus only on premature stop codon (PMSC) mutations, not any other type of mutations? If other interesting mutations in the inlA or other genes are obtained from the whole genome sequencing data, it would be better to list them as supplementary file.

2.    Line 91: Please elaborate ST on its first appearance in the manuscript.

3.   It is unclear why clinical isolates from Lombardy region and Lazio & Tuscany regions are presented separately in material and methods section. Are STs, CCs and presence of PMSCs were determined in cases from Lombardy region only (as mentioned in section 4.3.1, para 2, line 300) but not from Lazio and Tuscany regions? Please clarify.

4.    Table 2: Please provide details about the superscript mentioned in table 2 (ST1, CC2, PMSC3).

5. Figure 3 and line 114: It would be good to mention that ST9 appears to correlate with vegetable also, maybe to lesser extent than with meat.
